# Development of a New Ultra-High-Precision Magnetic Abrasive Finishing for Wire Material Using a Rotating Magnetic Field

**DOI:** 10.3390/ma12020312

**Published:** 2019-01-20

**Authors:** Lida Heng, Cheng Yin, Seok Ho Han, Jun Hee Song, Sang Don Mun

**Affiliations:** 1Division of Mechanical Design Engineering, Chonbuk National University, 664-14, Duckjin-gu, Jeonju 561-756, Korea; henglida1@gmail.com (L.H.); yoonsung_ys@hotmail.com (C.Y.); hahagggg@naver.com (S.H.H.); 2Division of Convergence Technology Engineering, Chonbuk National University, 664-14, Duckjin-gu, Jeonju 561-756, Korea

**Keywords:** ultra-high-precision magnetic abrasive finishing, AISI 1085 steel wire, rotating magnetic field, vibration frequency, surface roughness, removed diameter

## Abstract

In this paper, we propose a new ultra-high-precision magnetic abrasive finishing method for wire material which is considered to be difficult with the existing finishing process. The processing method uses a rotating magnetic field system with unbonded magnetic abrasive type. It is believed that this process can efficiently perform the ultra-high-precision finishing for producing a smooth surface finish and removing a diameter of wire material. For such a processing improvement, the following parameters are considered; rotational speed of rotating magnetic field, vibration frequency of wire material, and unbonded magnetic abrasive grain size. In order to evaluate the performance of the new finishing process for the wire material, the American Iron and Steel Institute (AISI) 1085 steel wire was used as the wire workpiece. The experimental results showed that the original surface roughness of AISI 1085 steel wire was enhanced from 0.25 µm to 0.02 µm for 60 s at 800 rpm of rotational speed. Also, the performance of the removed diameter was excellent. As the result, a new ultra-high-precision magnetic abrasive finishing using a rotating magnetic field with unbonded magnetic abrasive type could be successfully adopted for improving the surface roughness and removing the diameter of AISI 1085 steel wire material.

## 1. Introduction

Wire materials have been widely applied in many industries such as the music industry, medical industry, and mechanical industry, etc. In the medical industry, they can be used for pacemakers, active catheters, coronary stents, medical staples, orthodontics, implantations, and functional electrical stimulation systems [1,2,3,4,5,6,7,8,9]. For use in the medical industry, a smooth surface of these products is demanded to prevent bacteria adhesion on surfaces, surface contamination or corrosion, and to reduce the frictional features of the products [10]. Denstedt et al. [11] reported that a surface texture and a surface unevenness have real effects on the biocompatibility in a ureteral stent. He reported that the physical properties of biomaterials used for medical devices play an important role in biocompatibility because a device with a smooth surface can decrease the mechanical trauma at the device-tissue interface. The surface topographic characterizations of orthodontic arch wires significantly influence its biological and clinical behavior. However, it is very complicated to remove the surface irregularity or the diameter of these products by conventional finishing methods. For many years, some conventional methods have been successfully used for achieving the high quality of surface finish, namely filing, polishing, grinding, lapping, honing, burnishing and super finishing [12,13,14,15,16,17,18,19]. However, despite their great potential and their advantageous features, there are several limitations related to the surface accuracy and dimensional accuracy achievable by these conventional methods [20]. In particular, when the finishing material is a wire material or a micro-scale diameter material, the adoption of traditional methods such as filing, grinding, lapping, honing and super finishing is difficult to achieve because they require high pressures that can damage the surface that needs to be finished [21,22]. Rampal et al. [23] showed that traditional finishing methods such as lapping cannot enhance the surface quality of vacuum tubes, waveguides, and sanitary tubes, because of their shapes. Luo et al. [24] showed that the traditional finishing methods (i.e., grinding, lapping, and honing) cannot process and finish the surface of complex shape products, micro-scale sizes, and 3D structures efficiently. 

Recently, some advanced surface treatment methods (i.e., plasma coating and ion implantation) are being employed for achieving the good surface quality of these wire products instead of traditional methods [25,26,27,28,29]. Plasma coating is a method used in plasma, which is a highly reactive gas phase. In this method, the electrical discharge or microwave excitation is used to generate the plasma. Highly reactive species such as molecules in excited state, atoms, molecule fragments or even radicals initiate reactions resulting in good adherent and chemically stable surface layers [30]. An ion implantation is a method, which the nitrogen ions (N_3_^+^) or oxygen ions (O_2_^−^) are accelerated by an electrical field to be inserted into another solid material. In recent years, these surface treatment methods have been already used to enhance the physical properties of wire materials [31,32]. Krishnan et al. [10] reduced the surface roughness of nickel-titanium (Ni-Ti) wire and beta titanium (TMA) wire by using ion implantation method. They reported that the ion implantation can drastically improve the surface quality of Ni-Ti wire and TMA wire and it was able to reduce the original surface roughness of Ni-Ti wire from 795.95 nm to 330.87 nm, and the surface roughness of TMA wire from 463.28 nm to 236.35 nm. Chowdhury et al. [33] reported that the coating method is able to achieve the admirable surface finish and increase hardness simultaneously, and thus decreases the friction and wear rate.

However, despite significant advances of these methods, some disadvantages and limitations still exist. Rahman et al. [34] reported that the main risks from the plasma coating process are the electrical safety, infrared (IR) radiation, and ultraviolet (UV) radiation. In this process, there is a potential electrical shock hazard, which used a high process voltage is applied between the electrodes to produce the plasma. Because the ultraviolet (UV) radiation and infrared (IR) radiation are produced by the arc or plasma, so these radiations can pose a potential health hazard to the skin and the eyes. Ching et al. [35] reported that one common disadvantage of plasma coating is their adhesion to the workpiece, which allows the chemical interactions bonds between the layers. Additionally, the capabilities of coating in a cyclic loading condition are still under investigation.

The disadvantages of the ion implantation method can be stated as follows [36,37]:It requires very expensive equipment.It can cause damage in the substrate.It uses strong toxic gas sources (i.e., phosphine, and arsine).It has higher impurity content than diffusion.Difficult to achieve very deep and very shallow profiles.

Therefore, in order to overcome these disadvantages, new potential research related to such finishing processes is required. The purpose of this research is to develop a new ultra-high-precision magnetic abrasive finishing process, which can achieve the accuracy working of wire material with an admirable surface finish. In this study, a new ultra-high-precision magnetic abrasive finishing for wire material is developed using a rotating magnetic field system with unbonded magnetic abrasive type. A processing method depends on both a rotational motion of magnetic field and vibration of AISI 1085 steel wire workpiece (i.e., frequency and amplitude). Both parameters (e.g., rotational motion and vibration) play an important role in this work, as without vibration and with only rotational motion, circumferential grooves will form on the surface finish. 

Further the work aims at understanding the finishing characteristics of a new finishing process and to investigate the influence of critical process parameters. The main experimental parameters are the rotational speed of magnetic field, vibration frequency of wire workpiece, and unbonded magnetic abrasive grain size on change in surface roughness and removed diameter of AISI 1085 steel wire material.

## 2. Experimental Methods

Figure 1 schematically shows a magnetic force acting on a ferromagnetic particle in ultra-high -precision magnetic abrasive finishing process of AISI 1085 steel wire material. At position “A”, the magnetic force (*F_m_*) acting on a ferromagnetic particle, which is known as the finishing force and it is the sum of two forces, a force (*F_x_*) on x-direction and a force (*F_y_*) on y-direction. Both of forces on x-direction and y-direction play an important role during the finishing process and they are generated when the unbonded magnetic abrasive type is added to the finishing gap. A force (*F_x_*) is generated by the equipotential line and it acts on the ferromagnetic particle along the direction of the equipotential line. A force (*F_y_*) is generated by magnetic force line when the AISI 1085 steel wire workpiece pushes out the bridges formed in the direction of magnetic equipotential lines. Some of the force (*F_x_*) and (*F_y_*) can be calculated with reference to the literature [38,39,40,41,42].
(1)$$F_{m}=F_{x}+F_{y}$$
(2)$$F_{x}=\chi_{FP}\mu_{o}VH\frac{dH}{dX}\;\&\;F_{y}=\chi_{FP}\mu_{o}VH\frac{dH}{dY}$$
where $\chi_{FP}$ is the susceptibility of the ferromagnetic particles, $\mu_{o}$ is the permeability of free space, V is the volume of the ferromagnetic particles, H is the magnetic field intensity at point “A”, H=−∇φ, and dH/dX, dH/dY are the gradients magnetic field intensity in *x* and *y* directions.

Figure 2 shows a schematic diagram of the processing principle for an ultra-high-precision magnetic abrasive finishing of AISI 1085 steel wire material during the finishing process. The workpiece was inserted inside the finishing gap and vibrated by an electric slider, which enables a wire workpiece to vibrate up to 10 Hz and 2 mm of amplitude. The compound unbonded magnetic abrasive type (electrolytic iron particles, diamond abrasive particles and grinding oil) was used for the abrasive tools, and was supplied on the both edges of magnetic poles, which strongly surrounded the workpiece surface. They were magnetically joined together in the form of flexible magnetic abrasive brush (FMAB) between the magnetic poles (S and N) along the magnetic force lines. During the finishing process, FMAB gave a brush exhibiting relative rotational motion versus the vibrated workpiece surface, thus removing the unevenness from a wire material surface in the form of micro-chips.

## 3. Experiment of Ultra-High-Precision Magnetic Abrasive Finishing 

### 3.1. Design of Rotating Magnetic Field

Figure 3 shows a rotating magnetic field design used in the ultra-high-precision magnetic abrasive finishing process. The rotating magnetic field part is composed of two sets of Nd-Fe-B type permanent magnets (magnet dimensions: 20 mm × 10mm × 5 mm), two magnetic poles, a 1018 steel yoke, and a plastic chuck. The required magnetic field in the finishing zone is produced by Nd-Fe-B type permanent magnets attached to 1018 steel yoke inside the plastic chuck. In the ultra-high-precision magnetic abrasive finishing process, the high magnetic flux density can improve the finishing capabilities of the process. Thus, a 1018 steel yoke is specially designed for producing a high magnetic flux density inside the finishing zone. The result of magnetic flux density was measured by tesla meter (TM-601) (Kanetec, Chiyoda-Ku Tokyo, Japan) and the high magnetic flux density was found at both edges of magnetic poles inside the finishing zone is approximately 460 mT. Figure 3a shows a 2D detailed sketch with dimension of rotating magnetic field. The arrangement of two magnetic poles (S and N) with two sets of Nd-Fe-B type permanent magnets, which attached to a 1018 steel yoke is shown in Figure 3b.

### 3.2. Experimental Setup

The experimental setup of the ultra-high-precision magnetic abrasive finishing process was designed in this study (see Figure 4). The apparatus setup of this process is composed of two main systems: (i) a magnetic field rotating system, and (ii) a vibrating system. The most important rotating magnetic field is composed of a plastic chuck, a 1018 steel yoke, two sets of Nd-Fe-B type permanent magnets, and two magnetic poles. The speed control motor is used to generate the rotational motion of a rotating magnetic field (speed range: 90–1200 rpm). The vibrating system is composed of a computer, an electric slider, a slider controller, a programmable controller, a motion controller, an aluminum base, two aluminum arms, and an AISI 1085 steel wire. The AISI 1085 steel wire was inserted inside the gap of magnetic poles and both ends of the wire were connected to the aluminum arms of electric slider. The electric slider used in this vibrating system enables it to vibrate an AISI 1085 steel wire up to 10 Hz.

### 3.3. Material and Experimental Conditions

In order to evaluate the finishing capabilities for application to ultra-high-precision finishing of wire material, the AISI 1085 steel wires were selected as the workpiece with the size of (material dimensions: Ø 0.6 mm × (L) 280 mm) and its original surface roughness *Ra* before finishing was 0.25 µm. Table 1 and Table 2 show prescribed mechanical properties and chemical composition of the workpiece, respectively. Both types of accuracy of the workpiece (i.e., surface roughness and removed diameter) were improved by an ultra-high-precision magnetic abrasive finishing process at different rotational speeds of rotating magnetic field (350, 600, and 800 rpm). The detail of experimental conditions is shown in Table 3. The unbonded magnetic abrasive type has been selected as the abrasive tools for this study because it employs no such difficult preparation, and also provides good results with surface finish [43,44,45]. The unbonded magnetic abrasives type consists of 0.3 g diamond abrasive particle with the different mean diameters of (0.5, 3, and 6 µm), 0.8 g of electrolytic iron particles with the mean diameter of (Fe# 200 µm), and 200 µL of grinding oil. In order to improve the finishing efficiency, different vibration frequencies of AISI 1085 steel wire workpiece (4, and 10 Hz) and 2 mm of amplitude were applied to the process and finished by an ultra-high-precision magnetic abrasive finishing process for 120 s. According to the vibration frequencies at (4, and 10 Hz) and 2 mm of amplitude, the ultra-high-precision magnetic abrasive finishing process can finish the surface of AISI 1085 steel wire workpiece with 16 mm in length for 60 s. In order to understand the variations of surface roughness after the finishing process, the surface roughness parameter Ra (average surface roughness) were measured every 60 s using surface roughness tester (Mitutoyo SJ-400) (Mitutoyo, Sakado, Japan). Also, in order to understand the variations of the removed diameter (RD), the diameter of wire workpieces after finishing were measured every 60 s using a laser scan micrometer (LSM-6200) (Mitutoyo, Sakado, Japan). A scanning electron microscope (SEM at 100×, and 500×) (Hitachi, Tokyo, Japan) was also obtained to evaluate the effect of new ultra-high-precision magnetic abrasive finishing on the change in surface roughness of the AISI 1085 steel wire workpiece.

## 4. Results and Discussions

### 4.1. Effect of Rotational Speed of Rotating Magnetic Field on Finishing Process

In order to find the most favorable rotational speed of rotating magnetic field in terms of the surface roughness and removed diameter, the AISI 1085 steel wire workpieces were improved at different rotational speeds (350, 600, and 800 rpm) with 0.5 μm of magnetic abrasive grain size and 10 Hz of wire workpiece vibration. Figure 5 shows the effect of rotational speed of rotating magnetic field on surface roughness and the removed diameter. Three different rotational speeds have a significant effect on the improvement of surface roughness and the removed diameter of the wire workpiece (see Figure 5). In terms of the surface roughness, the rotational speed at 800 rpm is found to be the best for improving the surface roughness of wire workpiece and the original surface roughness was enhanced from 0.25 μm to 0.02 μm by high rotational speed at 800 rpm for 60 s. But after 60 s the surface roughness of wire workpiece did not enhance further, because the uneven surface was already removed after 60 s, and thus the ultra-high-precision magnetic abrasive finishing could not improve the surface roughness of wire workpiece anymore. The surface roughness slightly increased from 0.02 μm at 60 s to 0.03 μm at 120 s (see Figure 5). This can be explained due to the effect of removed chips on the finishing process after 60 s. The chips could not circulate well and remained on the surface of wire workpiece, and the remaining chips were refinished by the ultra-high-precision magnetic abrasive finishing. That’s why the surface roughness at 120 s became worse than the surface roughness at 60 s. However, the low rotational speed of rotating magnetic field is found to be the lowest improvement in surface roughness of wire workpiece. The original surface roughness was enhanced from 0.25 μm to 0.05 μm by low rotational speed at 350 rpm for 60 s. Also, in terms of the removed diameter, the high rotational speed at 800 rpm is found to be the best for removing the diameter of wire workpiece. The removed diameter at 350, 600, and 800 rpm was 3.97 μm, 5.24 μm, and 5.48 μm, respectively. Thus, it can be considered that the improvement of surface roughness and the removed diameter increase with the increase of rotational speed of the rotating magnetic field under the given experimental conditions. This can be explained due to the relationship between the centrifugal force and rotational speed of rotating magnetic field. In experimental conditions, it is expected that the highest rotational speed of 800 rpm produces highest centrifugal force, and thus creates more opportunities for improving the high quality of surface finish and more improvement of removed diameter. Hence, a rotational speed of rotating magnetic field at 800 rpm has been fixed for further experiments.

### 4.2. Effect of Unbonded Magnetic Abrasive Grain Size on Finishing Process

In order to find the most favorable abrasive grain size of magnetic abrasive, the wire workpieces were improved by different magnetic abrasive grain sizes (0.5, 3, and 6 μm) with an optimal rotational speed at 800 rpm and 10 Hz of wire workpiece vibration for 120 s. Figure 6 shows the effect of unbonded magnetic abrasive grain size on surface roughness and removed diameter. It can be seen from Figure 6 that the surface roughness and removed diameter of wire workpiece was improved with all the given conditions. The most significant improvement in surface roughness and removed diameter is observed when 0.5 μm of magnetic abrasive grain size was used. The original surface roughness of wire workpiece was enhanced from 0.25 μm to 0.02 μm. This can be explained due to the relationship between the small grain size of magnetic abrasive and high centrifugal force. When the small grain size of magnetic abrasive (0.5 μm) was used, the high centrifugal force tends to push the abrasive particles strongly against the wire workpiece surface. But after 60 s the surface roughness of wire workpiece did not enhance further and the surface roughness slightly increased between 60 s and 120 s. This can be explained due to the effect of removed chips on the finishing process after 60 s. Conversely, the original surface roughness is difficult to enhance to 0.06 μm, when the grain size of magnetic abrasive bigger than 3 μm. Also, in terms of the removed diameter, the highest improvement of removed diameter is observed when 0.5 μm was used. The removed diameter with 0.5 μm, 3 μm, and 6 μm of magnetic abrasive grain size was 5.48 μm, 3.69 μm, and 2.26 μm, respectively. Hence, 0.5 μm of abrasive grain size has been fixed for further experiments.

### 4.3. Effect of Wire Workpiece Vibration Frequency on Finishing Process

In order to find the most significant vibration frequency in improvement of surface roughness and removed diameter, the wire workpieces were finished by different wire workpiece vibration frequencies (4, and 10 Hz) with optimal rotational speed at 800 rpm and optimal magnetic abrasive grain size (0.5 μm) for 120 s. Figure 7 shows the effect of vibration frequency of wire workpiece on surface roughness and removed diameter. The improvement of surface roughness and removed diameter are increasing with the increase of vibration frequency (see Figure 7). In terms of the surface roughness, the highest improvement is observed when 10 Hz was used and the original surface roughness was enhanced from 0.25 μm to 0.02 μm for 60 s. However, 4 Hz is found to be the lowest improvement when compared to 10 Hz, and the original surface roughness was enhanced from 0.25 μm to 0.04 μm for 60 s. In terms of the removed diameter, the highest improvement of the removed diameter is observed when 10 Hz was used. However, when 4 Hz was used, the improvement of the removed diameter was generally smaller than 10 Hz of vibration frequency. The removed diameter in 120 s with 4 Hz, and 10 Hz was 5.1 μm, and 5.48 μm, respectively. Thus, it can be considered that the improvement of surface roughness and removed diameter increase with the increasing of vibration frequency of wire workpiece under the given conditions. This can be explained due to the relationship between the friction and vibration frequency of wire workpiece. The friction is generated when the wire workpiece is vibrated against the rotational motion of magnetic abrasives. Thus, when the vibration frequency increases up to 10 Hz, the friction between the surface of wire workpiece and magnetic abrasive decreases. Therefore, the decrease of friction creates more opportunities for improving the operational accuracy of the wire workpiece.

Figure 8 shows SEM micro pictures of the wire workpiece surfaces before and after processing by an ultra-high-precision magnetic abrasive finishing using a rotating magnetic field (SEM at 100×, and 500×). Figure 8a shows SEM micro pictures of the wire workpiece surface before finishing. Figure 8a has a roughness of 0.25 μm and the original scratches and irregular asperities can be found throughout the wire workpiece surface. Figure 8b shows SEM micro pictures of the wire workpiece surface after finishing for 60 s by 0.5 μm of magnetic abrasive grain size and 10 Hz of vibration frequency at 800 rpm. It can be seen from Figure 8b that the original scratches and irregular asperities were entirely removed from the surface of the wire workpiece, and surface roughness was 0.02 μm. Figure 8c,d shows SEM micro pictures of the wire workpiece after finishing for 60 s by 0.5 μm of magnetic abrasive grain size and 10 Hz of vibration frequency at 350 rpm, and 600 rpm, respectively. It can be seen that scratches and irregular asperities from the original surface were removed. However, the small scratches from the original surface remained and their surface roughness was 0.05 μm. Figure 8e,f shows SEM micro pictures of the wire workpiece surface after finishing for 60 s by 3 μm and 6 μm of magnetic abrasive grain size and 10 Hz of vibration frequency at 800 rpm. It can be seen from Figure 8e,f that the multiple scratches and irregular asperities from the original surface remained and their surface roughness was 0.14 μm. It is evident that the big grain sizes of magnetic abrasive did not show a significant effect on the surface improvement of wire workpiece. Figure 8g shows SEM micro pictures of the wire workpiece surface after finishing for 60 s by 4 Hz of vibration frequency. It can be seen that the scratches and irregular asperities were removed from the surface of wire workpiece and surface roughness was 0.04 μm. However, the surface finished by 4 Hz of vibration frequency in Figure 8g is not as smooth as the surface finished by 10 Hz in Figure 8b.

## 5. Conclusions

In this paper, we propose a new ultra-high-precision magnetic abrasive finishing for wire material using a rotating magnetic field. The results of this research can be summarized as follows:
The study revealed that a new development of ultra-high-precision magnetic abrasive finishing for wire material using a rotating magnetic field was successfully performed. This process could be used for improving the surface roughness and the removed diameter of wire material instead of some advanced surface treatment methods such as plasma coating, and ion implantation.This new ultra-high-precision magnetic abrasive finishing has several advantages over the conventional finishing processes. These advantages are that it has the capability for producing a smooth surface finish and removing a diameter of wire material, while the other finishing processes cannot achieve this.The highest rotational speed used in this study, which is 800 rpm, shows a significant effect on the improvement of surface roughness and removed diameter of wire material. This can be explained due to the high centrifugal forces generated by the highest rotational speed at 800 rpm.The highest improvement of surface roughness and removed diameter is observed when 0.5 μm of magnetic abrasive grain size was used with the increasing of the wire workpiece frequency at a certain level.The best result is found when the wire material workpiece finished with a 0.5 μm magnetic abrasive grain size and 10 Hz of vibration frequency at 800 rpm for 60 s and the original surface roughness was enhanced from 0.25 µm to 0.02 µm.Despite the potential advantages of the ultra-high-precision magnetic abrasive finishing process, the one disadvantage is that this finishing process can remove the diameter of the wire workpiece only in terms of the micro-scale removal.Current and future works are focused on the development of a new method for application to ultra-high-precision finishing of AISI 1085 steel wire material. According to the result, this method has significant potential for achieving a good surface finish and a removed diameter with wire material such as AISI 1085 steel wire with the size of 0.6 mm in diameter. In our future work, this method will apply to ultra-high-precision finishing of the biomaterials with a diameter smaller than 0.6 mm (i.e., nickel titanium wire, 316L stainless steel wire, and titanium molybdenum alloy arch wire).

## Figures and Tables

**Figure 1 materials-12-00312-f001:**
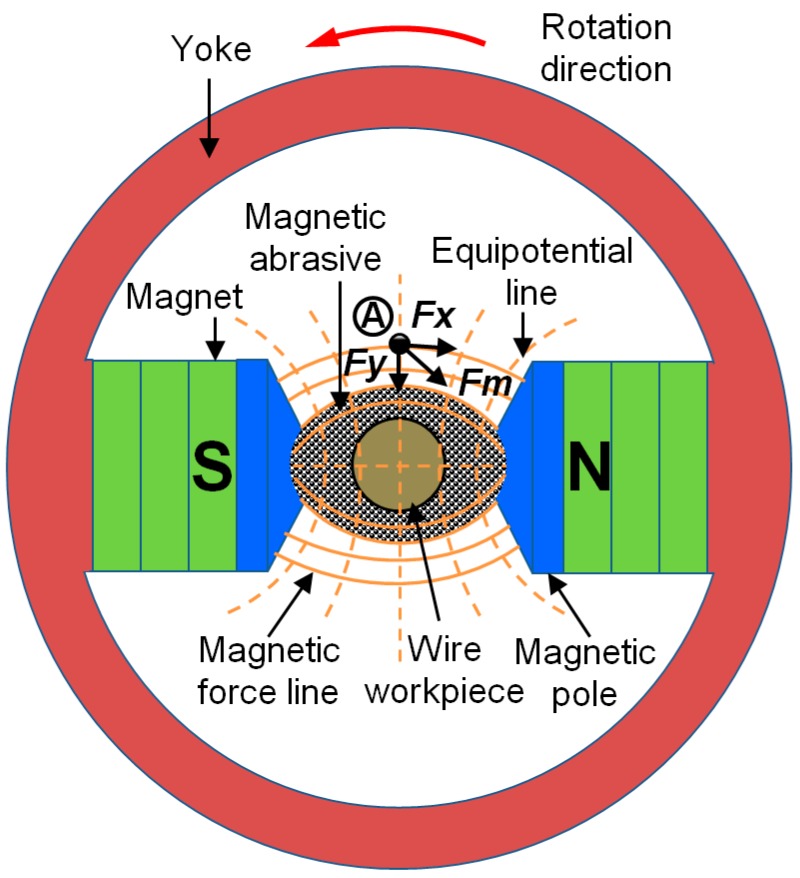
Schematic of processing principle for an ultra-high-precision magnetic abrasive finishing.

**Figure 2 materials-12-00312-f002:**
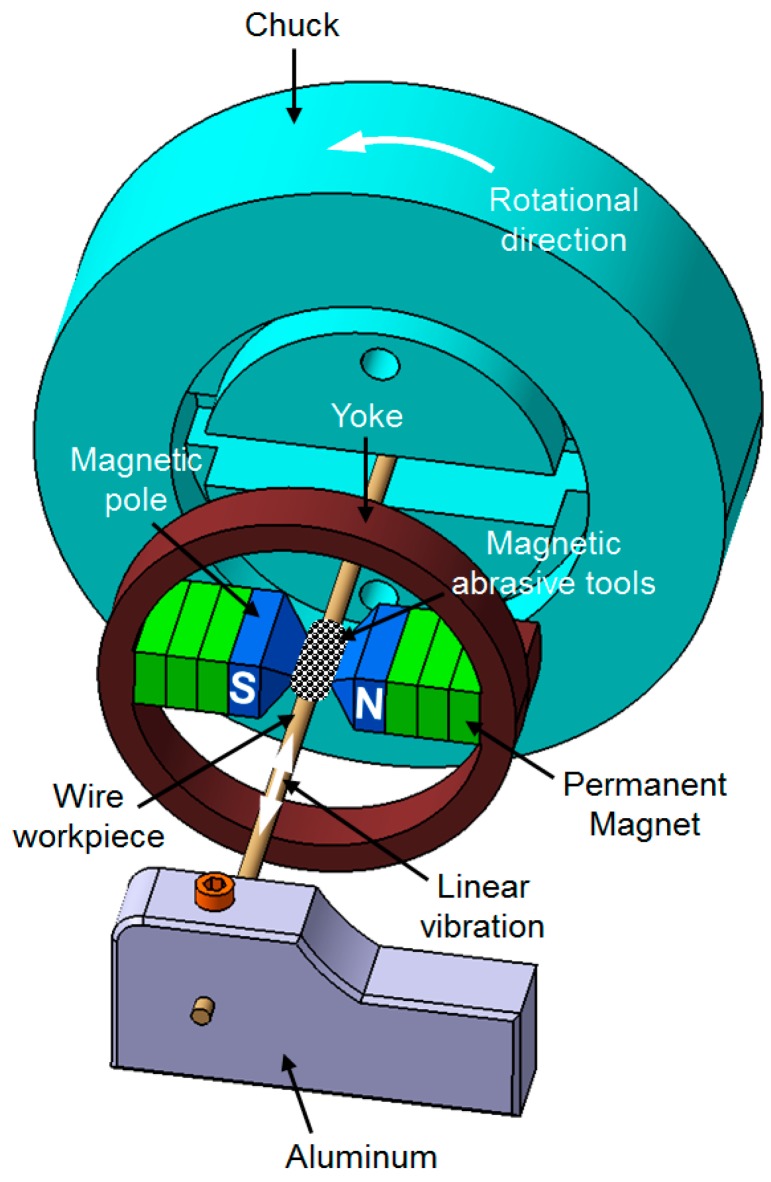
Schematic diagram of ultra-high-precision magnetic abrasive finishing for wire material using a rotating magnetic field.

**Figure 3 materials-12-00312-f003:**
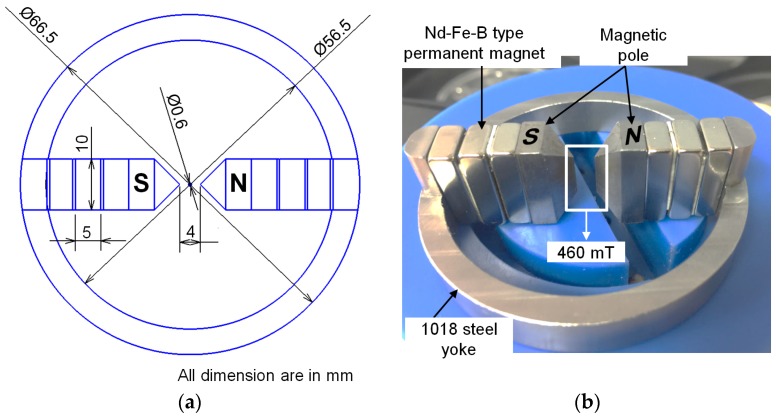
Design of rotating magnetic field used in an ultra-high-precision magnetic abrasive finishing. (**a**) 2D detailed sketch with dimension of rotating magnetic field (**b**) Photograph of rotating magnetic field without an unbonded magnetic abrasive type.

**Figure 4 materials-12-00312-f004:**
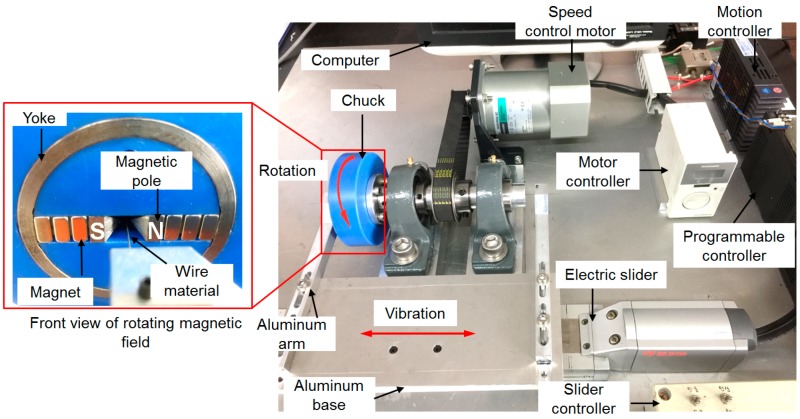
Experimental setup of ultra-high-precision magnetic abrasive finishing process.

**Figure 5 materials-12-00312-f005:**
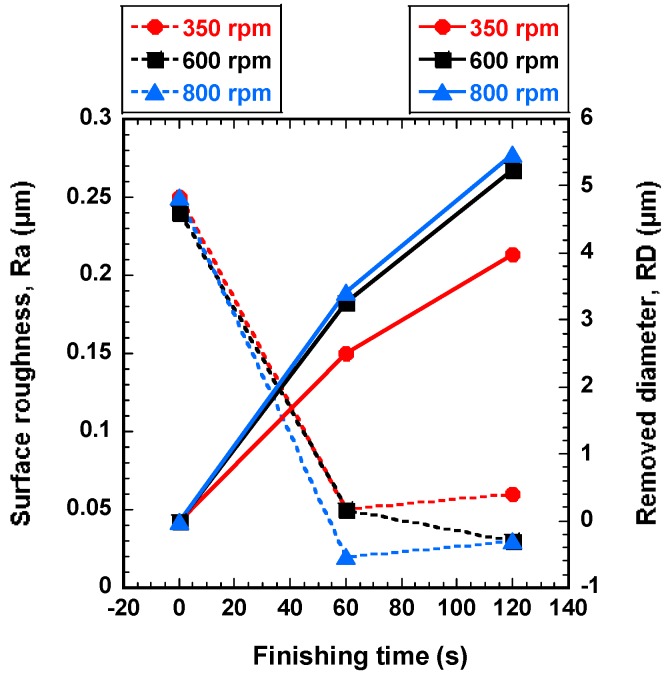
Effect of rotational speed of rotating magnetic field on surface roughness and the removed diameter (abrasive: 0.5 μm, 10 Hz).

**Figure 6 materials-12-00312-f006:**
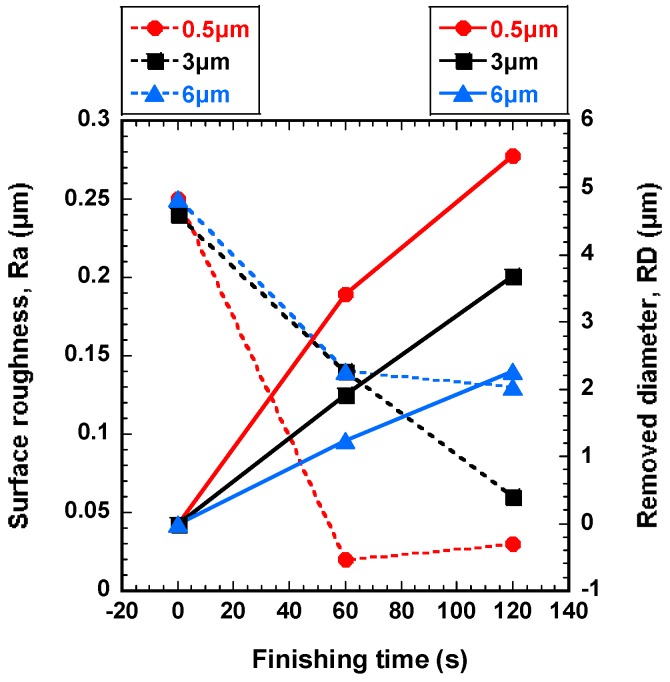
Effect of unbonded magnetic abrasive grain size on surface roughness and the removed diameter (rotational speed: 800 rpm, 10 Hz).

**Figure 7 materials-12-00312-f007:**
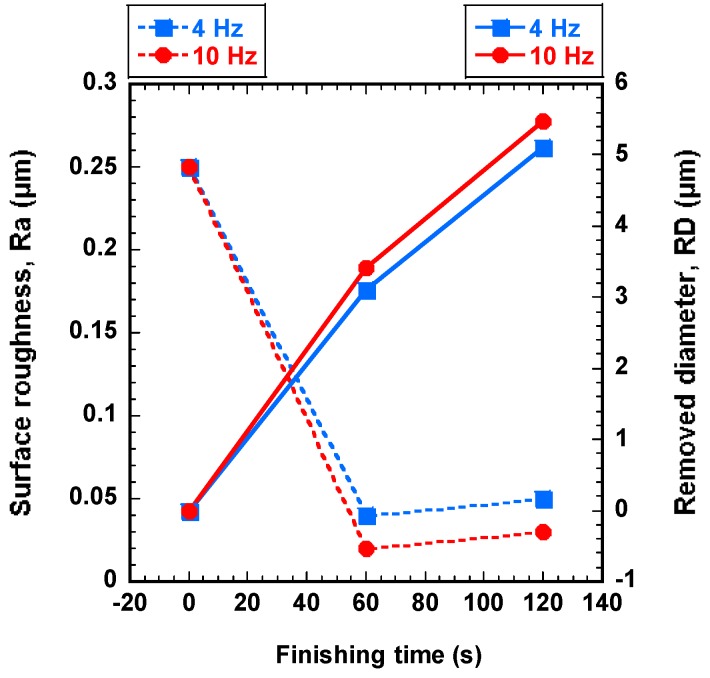
Effect of vibration frequency of wire workpiece on surface roughness and the removed diameter (abrasive: 0.5 μm, rotational speed: 800 rpm).

**Figure 8 materials-12-00312-f008:**
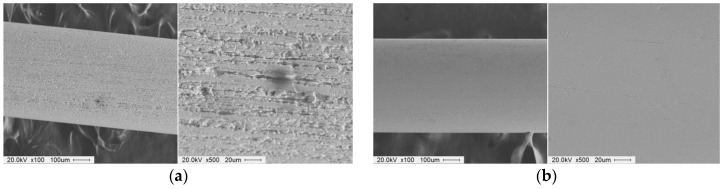
SEM micro pictures of the wire workpiece surface before and after processing by an ultra-high-precision magnetic abrasive finishing using a rotating magnetic field (SEM at 100× and 500×). (**a**) Before finishing (Ra: 0.25 μm); (**b**) After finishing (Ra: 0.02 μm, magnetic abrasive: 0.5 μm, rotational speed: 800 rpm, 10 Hz, 60 s); (**c**) After finishing (Ra: 0.05 μm, magnetic abrasive: 0.5 μm, rotational speed: 350 rpm, 10 Hz, 60 s); (**d**) After finishing (Ra: 0.05 μm, magnetic abrasive: 0.5 μm, rotational speed: 600 rpm, 10 Hz, 60 s); (**e**) After finishing (Ra: 0.14 μm, magnetic abrasive: 3 μm, rotational speed: 800 rpm, 10 Hz, 60 s); (**f**) After finishing (Ra: 0.14 μm, magnetic abrasive: 6 μm, rotational speed: 800 rpm, 10 Hz, 60 s); (**g**) After finishing (Ra: 0.04 μm, magnetic abrasive: 0.5 μm, rotational speed: 800 rpm, 4 Hz, 60 s).

**Table 1 materials-12-00312-t001:** Prescribed mechanical properties of AISI 1085 steel wire.

Tensile Strength	Yield Strength	Hardness	Elasticity	Fatigue Resistance
2709 MPa	537 MPa	58–63 HRc	190–210 GPa	10^7^ cycles

**Table 2 materials-12-00312-t002:** Prescribed chemical composition of AISI 1085 steel wire.

Iron	Carbon	Manganese	Phosphorus	Sulfur
97.98–98.41%	0.8–0.93%	0.7–1.0%	0.04% max	0.05% max

**Table 3 materials-12-00312-t003:** Experimental conditions.

Input Parameter	Value Standard Units
Wire workpiece	AISI 1085 steel wire(Size: Ø 0.6 mm × (L) 280 mm)
Rotational speed of rotating magnetic field	350, 600, 800 rpm
Electrolytic iron particle	0.8 g (200 μm mean diameter)
Diamond abrasive particle (PCD)	0.3 g (0.5, 3, 6 µm mean diameter)
Grinding oil	200 µL (light oil)
Magnetic pole geometry	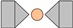
Magnetic flux densityin finishing zone	460 mT
Vibration frequency of wire workpiece	4, 10 Hz, and Amplitude: 2 mm
Finishing time	0 s, 60 s, 120 s

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
