# Peer review of "Development of a New Ultra-High-Precision Magnetic Abrasive Finishing for Wire Material Using a Rotating Magnetic Field"

_materials, 2019, doi:10.3390/ma12020312_

Reviewer 1 Report

It would be good to emphasize throughout the paper what makes this finishing process new and different.

Author Response

We would like to thank the editor and reviewers for careful and thorough reading of this manuscript and for the thoughtful comments and constructive suggestions, which help to improve the quality of this manuscript.

Reviewer 2 Report

The manuscript deals with the magnetic abrasive finishing for wire material using a rotating magnetic field. The manuscript is well written and presented well. However, very few English checks required.

1.     on page.5, lines- 158,159, it is not clear please check the English of lines, "in magnetic field rotating system, a conducted motor in which can be adjusted by the motor controller".

2.     Authors identified that 10Hz vibration has better surface finishing, could you please specify, how much length of the wire can be finished in specified time? that means how fast we can do the work.

3.      In fig. 5, 6 and 7, it is understood that the dashed lines indicate the surface roughness. The data presented well and the data indication is clear. that is roughness with increasing rpm, but it is not clear that, for higher rotation speed (800 rpm), the surface roughness is slightly increased for 60 sec to 120 sec! similarly, the effect of unbonded magnetic abrasive grain size on surface roughness and removed diameter for 0.5 um the surface roughness is shown the increasing trend between 60 and 120. could you please explain in the text.

Author Response

(The authors gave the same response as above.)

Reviewer 3 Report

Dear Authors,

The article aim is to investigate the influence of process parameters (ultra-high-precision magnetic abrasive finishing) on the surface roughness and diameter of a 1085 steel wire.

The introduction provides a good theoretical framework for conventional finishing methods. 

However i would suggest to improve line 38: "with a good surface finish"

What is a good surface finish? Is there a range of roughness values, which is interesting? 

Why is  a reduced diameter  a improved diameter? Which diameter range is desired? 

Furthermore, i suggest to mention a disadvantage of plasma coating. 

Another suggestion is to shift line 89 - 96 to Materials and Methods part. 

There is a lack of literature and comparison to other  magnetic abrasive machining methods. What is new in the new development?

I suggest to add some processing information in Materials and Methods for the 1085 steel wire material.

I miss the surface roughness measurement method. There is no description. Furthermore i  suggest a short description of Ra with a small draft or at least a clear one sentence definiton.

In general i miss a short description of diameter, roughness and SEM measurements in the experimental part.

I miss a discussion with advantage and disadvantages ultra-high-precision magnetic abrasive finishing for wire material using a rotating magnetic field. Are the obtained surface roughness values and diameters comparable with values used in medical industry? 

I miss a general discussion including recent literature. 

Author Response

(The authors gave the same response as above.)
